# mRNAs and lncRNAs intrinsically form secondary structures with short end-to-end distances

Wan-Jung C. Lai[1], Mohammad Kayedkhordeh[1], Erica V. Cornell[1], Elie Farah[1], Stanislav Bellaousov[1], Robert Rietmeijer[1], Enea Salsi[1], David H. Mathews[1] & Dmitri N. Ermolenko[1]

The 5′ and 3′ termini of RNA play important roles in many cellular processes. Using Förster resonance energy transfer (FRET), we show that mRNAs and lncRNAs have an intrinsic propensity to fold in the absence of proteins into structures in which the 5′ end and 3′ end are ≤7 nm apart irrespective of mRNA length. Computational estimates suggest that the inherent proximity of the ends is a universal property of most mRNA and lncRNA sequences. Only guanosine-depleted RNA sequences with low sequence complexity are unstructured and exhibit end-to-end distances expected for the random coil conformation of RNA. While the biological implications remain to be explored, short end-to-end distances could facilitate the binding of protein factors that regulate translation initiation by bridging mRNA 5′ and 3′ ends. Furthermore, our studies provide the basis for measuring, computing and manipulating end-to-end distances and secondary structure in RNA in research and biotechnology.

[1] Department of Biochemistry & Biophysics and Center for RNA Biology, School of Medicine and Dentistry, University of Rochester, Rochester, NY 14642, USA. These authors contributed equally: Wan-Jung C. Lai, Mohammad Kayedkhordeh. Correspondence and requests for materials should be addressed to D.H.M. (email: David_Mathews@URMC.Rochester.edu) or to D.N.E. (email: Dmitri_Ermolenko@URMC.Rochester.edu)

The sequence and secondary structures at the 5′ and 3′ termini of RNA have important roles in various cellular processes[1–3]. RNA termini are recognized by a number of protein factors that mediate RNA processing and mRNA translation[4]. The binding of protein factors to RNA termini, rates of RNA degradation by exonucleases and translation regulation are affected by the intramolecular secondary structure of RNA[5–8].

Recent theoretical analyses of a number of randomized and natural RNA sequences suggested that the 5′ and 3′ ends of long (1000–10,000 nucleotide-long) RNAs are always brought in the proximity of few nanometers of each other regardless of RNA length and sequence because of the intrinsic propensity of RNA to form widespread intramolecular basepairing interactions[9–11]. One study predicted that the 5′ to 3′ end distance in RNAs is 3 nm, on average[10]. These theoretical predictions were tested by single-molecule Förster resonance energy transfer (smFRET) measurements of end-to-end distances in several viral RNAs and mRNAs from the fungus *Trichoderma atroviride*, which varied in length between 500 and 5000 nucleotides and were folded in vitro in the absence of any protein factors[12]. Experimentally derived end-to-end distances in RNA molecules, in which FRET was detected, ranged between 5 and 9 nanometers[12].

The intrinsic closeness of RNA ends may have a number of biological implications for various aspects of mRNA metabolism including translation, splicing or degradation. However, the hypothesis that the closeness of RNA ends is a universal property of all natural transcripts remains to be systematically tested. Previous computational studies explored end-to-end distances in a limited number of natural sequences[9–11]. The propensity of human mRNAs and lncRNAs to fold into structures with short end-to-end distances has also not been examined. In a published smFRET study[12], end-to-end distance was measured only in molecules that showed FRET, which might have represented only a minor fraction of the total population of RNA molecules. It is further unclear to what extent end-to-end distance may vary between different transcripts. Sequence features that define RNA potential to fold into structures with short end-to-end distances are unknown.

Here, we use FRET measurements and computational analysis of RNA structure to examine the end-to-end distances in mRNAs and lncRNAs from several species, including humans. Our study focuses on natural mRNA and lncRNA sequences because of their biological importance. We find that most, if not all, mRNAs and lncRNAs have an intrinsic propensity to fold into structures with short end-to-end distances irrespective of their length and sequence.

## Results

**The 5′ and 3′ ends of human mRNAs are intrinsically close**. We experimentally determined the end-to-end distance in a number of mRNAs using FRET between fluorophores introduced at each end of the mRNAs (Fig. 1a). The range of FRET sensitivity (1 to 10 nm for Cy3-Cy5 pair[13]) to distance changes matches the theoretically predicted array of distances between the 5′ and 3′ ends of structured RNAs[10,11]. We selected yeast and human mRNAs that encode abundant housekeeping proteins and have well-annotated 5′ and 3′ UTR sequences, such as yeast RPL41A (ribosomal protein L41A) and human GAPDH (glyceraldehyde-3-phosphate dehydrogenase) (Supplementary Table 1). In addition, we used rabbit β-globin and firefly luciferase (Fluc) mRNAs that have been used as canonical "standard" mRNAs in many previous mechanistic studies of eukaryotic translation.

We labeled the 5′ and 3′ ends of mRNAs, which lacked the 5′ cap and poly(A) tail, with donor (Cy3) and acceptor (Cy5) fluorescent dyes, respectively. Computational prediction of RNA secondary structure suggests that all examined mRNAs can form extensive intramolecular basepairing interactions (Fig. 1a, Supplementary Fig. 1a, b). mRNAs were refolded in the absence of protein factors and the presence of 100 mM KCl and 1 mM MgCl₂. These ionic conditions are considered to be optimal for translation in eukaryotic in vitro translation systems[14,15]. Furthermore, the 1 mM concentration of Mg²⁺ used in our experiments is similar to concentrations of free (unbound) cytoplasmic Mg²⁺ in human cells (0.5–1 mM)[16]. Energy transfer between fluorophores attached to the 5′ end of the 5′ UTR and 3′ end of the 3′ UTR was detected in all eight tested mRNAs. The average end-to-end distances, which were determined for each transcript from ensemble FRET data, were in the range of 5–7 nm irrespective of mRNA length (Fig. 1b, Supplementary Table 1). These distances are two to ten times shorter than those predicted for unstructured RNA by the freely jointed chain model[17,18] of polymer theory (Fig. 1b).

We next tested whether mRNA ends are brought into close proximity by basepairing interactions. Refolding of human GAPDH and rabbit β-globin mRNAs in the presence of a 50-nucleotide-long DNA oligonucleotide complementary to the 3′ end of the respective mRNA led to a marked reduction in the efficiency of energy transfer between fluorophores attached to RNA ends (Fig. 1c). The observed decrease in FRET efficiency is presumably due to annealing of the DNA oligonucleotide to the 3′ end of the mRNA and disruption of the intramolecular secondary structure (Supplementary Fig. 2a, b).

To further test the effect of intramolecular secondary structure on mRNA end-to-end distance, we replaced 106 nucleotides at the 5′ end of the 116 nucleotide-long 5′ UTR of GAPDH mRNAs with 53 CA repeats, which have low basepairing potential, to create the 5′UTR(CA)₅₃GAPDH mRNA variant. Likewise, 53 CA repeats were inserted at the 3′ end of the 202 nucleotide-long 3′ UTR of GAPDH mRNA in place of 106 nucleotides of the original sequence, to make the 3′UTR(CA)₅₃GAPDH mRNA variant (Supplementary Fig. 2c). No energy transfer between the mRNA ends was detected in either of the GAPDH mRNA variants containing CA repeats, i.e. in 5′UTR(CA)₅₃GAPDH and 3′UTR(CA)₅₃GAPDH mRNAs (Fig. 1c). These results indicate that the 5′ and 3′ ends of wild-type GAPDH mRNA were brought within FRET distance via the formation of intramolecular basepairing interactions.

mRNAs in eukaryotic cells undergo 5′ capping (attachment of 7-methyl-guanosine to the 5′ end) and polyadenylation of the 3′ end. In the experiments described above, we measured the distance between the 5′ end of the 5′ UTR and the 3′ end of the 3′ UTR of model mRNAs in the absence of the 5′ cap and the poly (A) tail because neither the 5′ cap nor adenosine repeats are likely to significantly affect secondary structure of mRNAs that lack extended uridine repeats. We tested the validity of this assumption by attaching donor and acceptor fluorophores to the ends of β-globin and GAPDH mRNAs transcribed with a 30 nucleotide-long poly(A) tail. Addition of a poly(A) tail led to a significant reduction in FRET efficiency (Fig. 1c), corresponding to an increase of end-to-end mRNA distance in both GAPDH and β-globin mRNAs by ~5 nm (Supplementary Table 1). This value is consistent with the ~5 nm end-to-end distance predicted for the 30-nt-long RNA segment in the random-coil conformation[19]. Therefore, the poly(A) tail is unstructured and not involved in basepairing interactions with the 5′ UTR.

**mRNAs fold into a dynamic ensemble of structures**. Published studies[20–22] and our own computational predictions suggest that mRNAs fold into an ensemble of structures with comparable thermodynamic stabilities rather than into a single structure. To

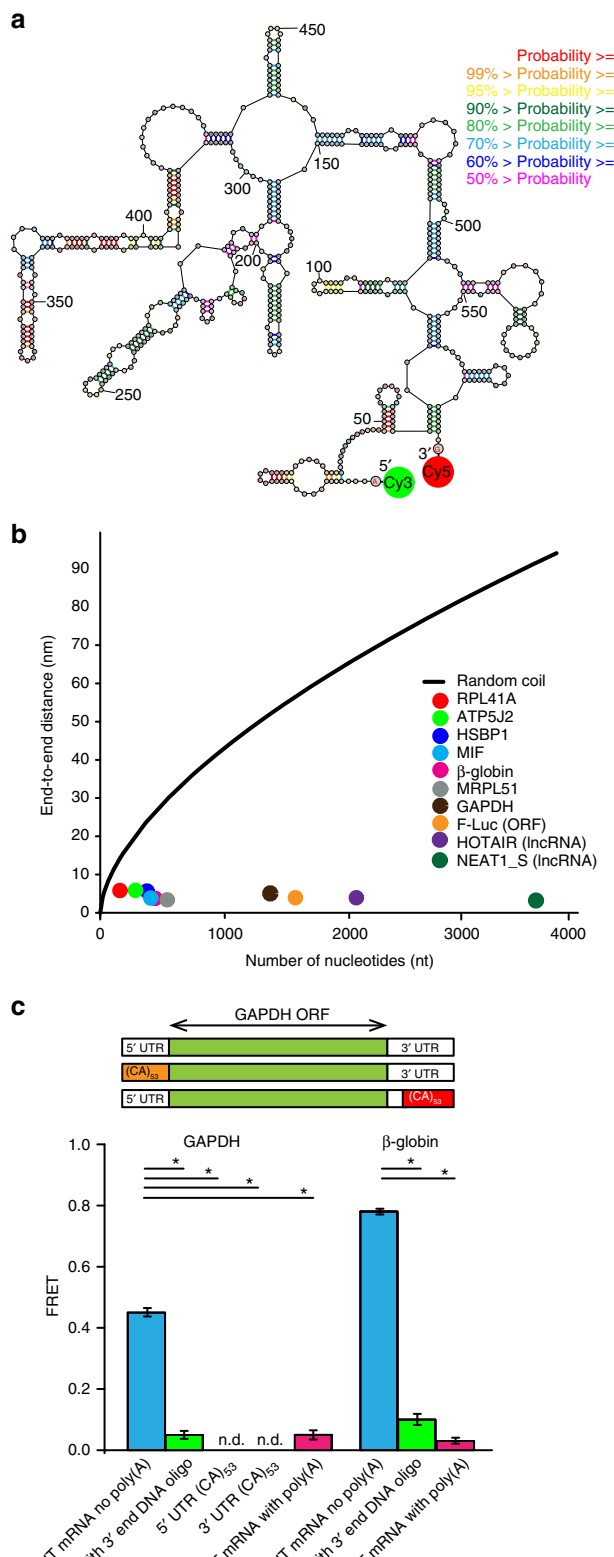

**Fig. 1** RNA ends are brought within FRET distance via the formation of intramolecular basepairing interactions. **a** An exemplary secondary structure from the ensemble of structures of rabbit β-globin mRNA lacking poly(A) tail predicted by free energy minimization. Base pair probabilities, predicted with a partition function, are color-coded. In order to measure the end-to-end distance by FRET, the 5′ and 3′ ends of mRNA were conjugated with donor (Cy3) and acceptor (Cy5) fluorophores, respectively, as indicated. **b** Average end-to-end distances of mRNAs and lncRNAs, which were folded in the presence of 100 mM KCl and 1 mM MgCl$_2$, were determined by ensemble FRET measurements and plotted as a function of RNA length: yeast RPL41A mRNA (red), firefly luciferase mRNA (orange), rabbit β-globin mRNA (magenta), human ATP5J2 mRNA (green), HSBP1 mRNA (indigo), MIF mRNA (blue), MRPL51 mRNA (gray), GAPDH mRNA (brown), HOTAIR lncRNA (purple), and NEAT1_S lncRNA (dark green). The black line shows theoretically predicted end-to-end distance of unstructured RNA, assuming a freely jointed chain model. **c** FRET values were measured between fluorophores attached to the 5′ and 3′ ends of the following GAPDH and β-globin mRNAs: mRNA lacking poly(A) tail (blue); mRNA lacking poly(A) tail folded in the presence of a 50-nucleotide-long DNA oligonucleotide complementary to the 3′ end of mRNA (green); mRNA with poly(A) tail (pink). FRET could not be detected (n.d.) in GAPDH variants, which lacked poly(A) tail and contained 53 CA repeats introduced into the 5′ or 3′ UTR. Each FRET value represents the mean ± standard deviation (SD) of three independent experiments. A star indicates that FRET values are significantly different, as p-values determined by the Student t-test were below 0.05

which examined end-to-end distances in long transcripts, did not provide insights into mRNA structural dynamics because smFRET was measured in freely diffusing molecules[12]. To immobilize mRNAs to the surface of the microscope slide and, thus, to extend the observation time, a 20-nucleotide-long DNA oligonucleotide conjugated to biotin was annealed in the middle of the RNA where it was computationally predicted to have a minimal effect on the overall secondary structure and end-to-end distance of the RNA[23] (Supplementary Fig. 1c, d). No significant decrease in energy transfer in ensemble FRET experiments was observed when Cy3/Cy5-labeled GAPDH, β-globin, and MIF mRNAs lacking poly(A) tail were folded in the presence of biotin-labeled DNA oligonucleotides (Supplementary Fig. 3). Hence, annealing of biotin-labeled DNA oligonucleotides did not affect end-to-end distance nor disrupt the overall RNA structure.

mRNAs were tethered to the surface of microscope slides coated with BSA-biotin/ neutravidin and then imaged by exciting the donor (Cy3) fluorescence with the green (532 nm) laser. smFRET traces acquired for GAPDH, β-globin and MIF mRNAs exhibited single-step photobleaching of both donor and acceptor fluorophores, indicating that we observe intramolecular rather than intermolecular energy transfer between mRNA ends (Supplementary Fig. 4). Because efficiencies of labeling of the 5′ end with Cy5 (100%) and the 3′ end of mRNA with Cy3 (~20–30%) markedly differed, we also imaged mRNAs by exciting the acceptor (Cy5) fluorescence with the red (642 nm) laser. Single-step Cy5 photobleaching was observed in 97–99% of single-molecule traces of GAPDH, β-globin, and MIF mRNAs, indicating that mRNA dimers or higher-order oligomers were essentially absent.

FRET distribution histograms, which were constructed by compiling 300–1200 smFRET traces, are best fit to a sum of four (GAPDH mRNA) or three (β-globin and MIF mRNAs) Gaussians (Fig. 2a–c). The distinct FRET peaks in distribution histograms correspond to different FRET states and, thus, different mRNA end-to-end distances. During run-off transcription, T7 RNA polymerase can add one, two or three non-

test this prediction, we examined end-to-end distance in individual GAPDH, β-globin, and MIF mRNA molecules by measuring smFRET using total internal reflection fluorescence (TIRF) microscopy. smFRET reveals the structural dynamics of individual molecules that are masked in ensemble (bulk) FRET measurements because of signal averaging in the heterogeneous and non-synchronized population[13]. A previous smFRET study,

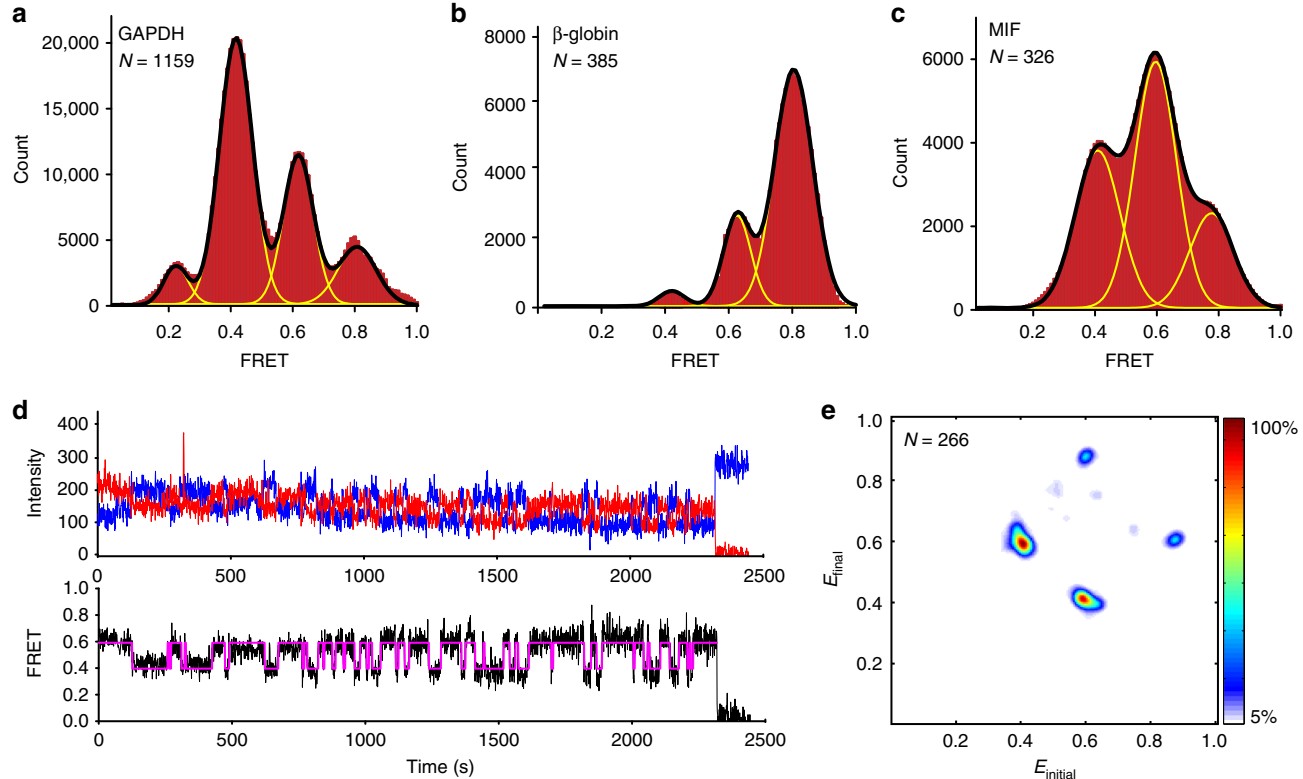

**Fig. 2** mRNAs fold into a dynamic ensemble of structures. smFRET was measured between the dyes attached to the 5′ end of the 5′ UTR and 3′ end of the 3′ UTR in GAPDH, β-globin, and MIF mRNAs. **a–c** Histograms, compiled from hundreds of smFRET traces, showing the distribution of the FRET values in **a** GAPDH mRNA, **b** β-globin mRNA, and **c** MIF mRNA folded in the presence of 100 mM KCl and 1 mM MgCl$_2$. Yellow lines represent individual Gaussian fits and black lines indicate the sum of Gaussians. N is the number of single-molecule traces compiled. **d** Representative smFRET trace for GAPDH mRNA showing fluctuations between the 0.4 and 0.6 FRET states. Observed intensities of donor and acceptor fluorescence and the calculated apparent FRET efficiency are shown in blue, red, and black, respectively. Hidden Markov Model fit is shown in magenta. **e** Transition density plot (TDP) analysis of 5114 fluctuations between different FRET states in 266 HMM-idealized FRET traces obtained for GAPDH mRNA. The frequency of transitions from the starting FRET value (x axis) to the ending FRET value (y axis) is represented by a heat map. The range of FRET efficiencies from 0 to 1 was separated into 200 bins. The resulting heat map was normalized to the most populated bin in the plot; the lower- and upper-bound thresholds were set to 5% and 100% of the most populated bin, respectively

templated nucleotides to the 3′ RNA end in a fraction of the transcripts[24]. To test whether the multiple FRET states in FRET histograms correspond to sequence or secondary structure heterogeneity, we varied the concentration of Mg$^{2+}$, which is known to stabilize the secondary and tertiary structure of RNA. An increase in MgCl$_2$ concentration during RNA folding from 0 to 8 mM raised ensemble FRET values in β-globin and GAPDH mRNAs (Supplementary Fig. 5). Consistent with ensemble FRET data, an increase in MgCl$_2$ concentration from 0 to 2 mM during mRNA refolding reduced the fraction of molecules exhibiting lower FRET states and increased the fraction of molecules showing higher FRET values without substantially altering the positions of the peaks of the FRET states (Supplementary Fig. 6) in GAPDH, β-globin, and MIF mRNAs. Hence, most, if not all, distinct FRET peaks of smFRET distribution histograms correspond to interconverting structural states. Nevertheless, a contribution of sequence heterogeneity to the breadth of smFRET histograms cannot be completely ruled out.

Consistent with the idea that mRNAs fold into a dynamic ensemble of several structural states with multiple end-to-end distances, individual smFRET traces in GAPDH, β-globin, and MIF mRNAs showed spontaneous fluctuations between distinct FRET states (Fig. 2d, Supplementary Fig. 4). Using GAPDH mRNA as an example, we further explored the statistics of fluctuations between FRET states via Hidden Markov Model (HHM) and Transition Density Plot analyses[25,26]. Consistent

with FRET distribution histograms, individual GAPDH mRNA molecules predominantly fluctuated between ~0.4, 0.6, and 0.8 FRET states at frequencies of ~0.1–0.03 s$^{-1}$ (Fig. 2e, Supplementary Table 2). These rates are similar to previously measured kinetics of the spontaneous transition between two alternative 5 base pair-long RNA helixes[27], indicating that FRET changes observed in GAPDH mRNA may correspond to analogous structural rearrangement. Although a few (1%) of the dynamic smFRET traces showed fluctuations between three (0.2, 0.4, and 0.8) or all four (0.2, 0.4, 0.6, and 0.8) FRET states (Supplementary Fig. 4), the majority (99%) of traces showed reversible fluctuations that transitioned between just two states (either between 0.4 and 0.6 or between 0.6 and 0.8; Fig. 2e). These data suggest there may be two dynamic structurally distinct sub-populations of GAPDH mRNA.

Interestingly, FRET between the 5′ and 3′ ends in β-globin and MIF mRNAs showed stronger dependence on Mg$^{2+}$ concentration than FRET in GAPDH mRNA (Supplementary Figs. 5, 6), suggesting that in addition to the formation of secondary structure, intramolecular tertiary interactions may contribute, albeit moderately, to bringing the ends of β-globin and MIF mRNAs in close proximity. Similar to the effect of magnesium ions, the addition of the molecular-crowding agent PEG-8000 to β-globin and MIF mRNAs produced a relatively small but nevertheless appreciable increase in FRET, indicating that mRNA structures with shorter end-to-end distances were stabilized,

while the average end-to-end distance in GAPDH mRNA was not affected (Supplementary Fig. 7). Molecular crowding is thought to promote RNA folding and the formation of more compact RNA conformations through the excluded volume effect[28–32]. The relatively small effect of molecular crowding on the end-to-end distance suggests that mRNAs do not fold into globular, highly condensed structures and further supports the idea that tertiary interactions have a minor role (if any) in bringing mRNA ends in close proximity.

**Computational estimates of end-to-end distances in RNA.** Using the RNAstructure software package[33] and a freely jointed chain polymer theory[19], we developed a new algorithm for modeling the distribution of end-to-end distances for the folding ensemble in natural RNAs to test the hypothesis about the proximity of RNA ends at a transcriptome-wide level. In this algorithm, a representative thermodynamic ensemble of structures is selected by stochastic sampling[34], and then the distance between the 5′ and 3′ ends is estimated for each member of the sample in nanometers. The calculation employs two segment sizes (unpaired nucleotides and helix ends), which are estimated based on the freely jointed chain model of polymer theory[17–19]. We do not consider the presence of the poly(A) tail of mRNAs because the poly(A) tail is unlikely to make base-pairing interactions with the rest of the mRNA (Fig. 1c). Thus, we estimate the distance between the 5′ end of mRNA and the 3′ end of the 3′ UTR at the junction with poly(A) tail. Our algorithm generates a histogram of estimated distances and, thus, examines both average end-to-end distance and the distribution of end-to-end distances in the population of RNA structures.

Average end-to-end distances derived from our ensemble FRET measurements correlate reasonably well with distances predicted for the same RNAs using computation with a linear regression coefficient, $r^2$, of 0.75 (Supplementary Table 1, Supplementary Fig. 8). Deviations between predicted and experimentally measured end-to-end distances do not exceed 3 nm and, at least in part, may result from perturbations of fluorescent properties of the donor and acceptor fluorophores due to local environmental effects, which may lead to a 0.5–1 nm error in the determination of FRET-derived distances[13,35]. Furthermore, FRET might overestimate the average end-to-end distance because a fraction of RNA may be misfolded or unfolded under chosen experimental conditions. Hence, our computational algorithm adequately predicts the end-to-end distance in the ensemble of folded RNA molecules and can be used to examine end-to-end distances in the human transcriptome.

**The ends are close in most mRNA and lncRNA sequences.** We used our algorithm to predict the end-to-end distance in 21,238 transcripts of the HeLa human cell transcriptome. The predicted end-to-end distances were relatively narrowly distributed with a population mean of ~4 nm (Fig. 3a). Only ~0.01% of all mRNAs were predicted to have end-to-end distances over 8 nm. Hence, the propensity of folding into structures with short end-to-end distances is common to all human mRNAs. Furthermore, closeness of mRNA ends appears to be largely independent of nucleotide sequence and mRNA length.

We further extended the analysis of computationally predicted end-to-end distances in the human transcriptome to lncRNAs. LncRNAs constitute a recently discovered class of non-coding cellular transcripts that are over 200 nucleotides in length. Similarly to mRNAs, the predicted end-to-end distances in 103,746 of human lncRNA sequences downloaded from the LNCipedia database[36] were relatively narrowly distributed with a population mean of ~4 nm (Fig. 3b). Only ~0.12 % of all lncRNAs were predicted to have end-to-end distances over 8 nm.

To test these computational predictions experimentally, we fluorescently labeled the 5′ and 3′ ends of HOTAIR and NEAT1_S lncRNAs. HOTAIR and NEAT1_S are two of the most well-studied lncRNAs, which are 2146 and 3734 nt-long, respectively. Changes in expression of HOTAIR and NEAT1_S lncRNAs disrupt normal embryonic development and promote tumorigenesis[37–39]. Chemical probing experiments showed that both HOTAIR and NEAT1_S lncRNAs form extensive secondary structures[37,39]. FRET between fluorophores attached to the 5′ and 3′ ends was detected in both HOTAIR and NEAT1_S lncRNAs folded in the presence of 100 mM KCl and 1 mM MgCl$_2$ (Fig. 3c). The average end-to-end distances in HOTAIR and NEAT1_S lncRNAs, which were determined from ensemble FRET data (Fig. 1b, Supplementary Table 1), are 10 and 20-fold shorter than those predicted for unstructured RNA of the same length. These experimental results validate our computational predictions suggesting that most, if not all, human lncRNA sequences have the propensity of folding into structures with short end-to-end distances.

To further explore the dependence of the end-to-end distance on RNA sequence, we estimated end-to-end distances in 10,000 variants of GAPDH mRNA, in which a segment of 106 nucleotides at 3′ end of the 3′ UTR was shuffled while preserving the original adenosine/guanosine/cytosine/uracil ratio (Fig. 4a). Similar to the distribution of end-to-end distances in the HeLa cell transcriptome, end-to-end distances in GAPDH variants with a shuffled sequence in the 3′ UTR were narrowly distributed with a population mean of ~3.8 nm (Fig. 4a). To experimentally test these computational estimates, we cloned, transcribed without the 3′ poly(A) tail and labeled with Cy3/Cy5 fluorophores one of the shuffled GAPDH variants ("Shuffled_1"). End-to-end distances for the original wild-type and Shuffled_1 GAPDH mRNAs were predicted to have equal end-to-end distances. Consistent with computational prediction, ensemble FRET values measured in wild-type and Shuffled_1 GAPDH mRNA variants were essentially indistinguishable (Fig. 4b, Supplementary Table 1).

We also examined how the end-to-end distance depends on GC content because G•C pairs are more stable than A•U pairs. To that end, we varied GC content between 0 and 100% with 5% increments in a 1327 nucleotide-long RNA (i.e., RNA equal in length to GAPDH mRNA). For each %GC value, we calculated average end-to-end distance over 100 sequences, each of which was generated randomly. The ratios A/U and G/C were fixed to the same value as those in GAPDH. Our computational analysis shows that variation in GC content does not substantially affect the end-to-end distance in RNA (Supplementary Fig. 9).

Taken together, our results provide evidence that RNA end-to-end distance is largely sequence independent. Hence, random RNA sequences tend to form secondary structure with short end-to-end distances. Worth mentioning, however, is that our computational analysis does not consider the formation of tertiary intramolecular interactions or pseudoknots, which might extend the end-to-end distance in some RNAs. One notable example of such structured RNA, in which the end-to-end distance is extended beyond 7 nm, is small subunit rRNA.

**Sequence properties of unstructured RNAs.** Although we find that the ends of most RNA sequences are inherently close, we have also demonstrated that the introduction of CA-repeats, which are known to have low basepairing potential, increase end-to-end distance in RNA. To further investigate the relationships between sequence properties, basepairing potential and end-to-end distance of RNA, we evolved the human GAPDH mRNA

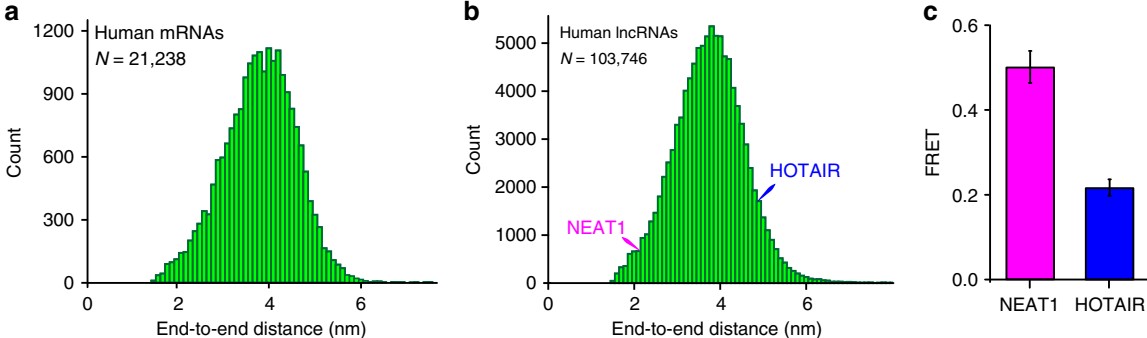

**Fig. 3** The ends of human mRNAs and lncRNAs are universally close. **a**, **b** Distribution of computationally predicted average end-to-end distances in **a** mRNA sequences from the HeLa cell transcriptome and **b** human lncRNA sequences. N is the number of sequences analyzed. Predicted end-to-end distances in NEAT1_S and HOTAIR lncRNAs are indicated by blue and pink arrows. **c** Ensemble FRET values measured in NEAT1_S and HOTAIR lncRNAs. Each FRET value represents the mean ± SD of three independent experiments

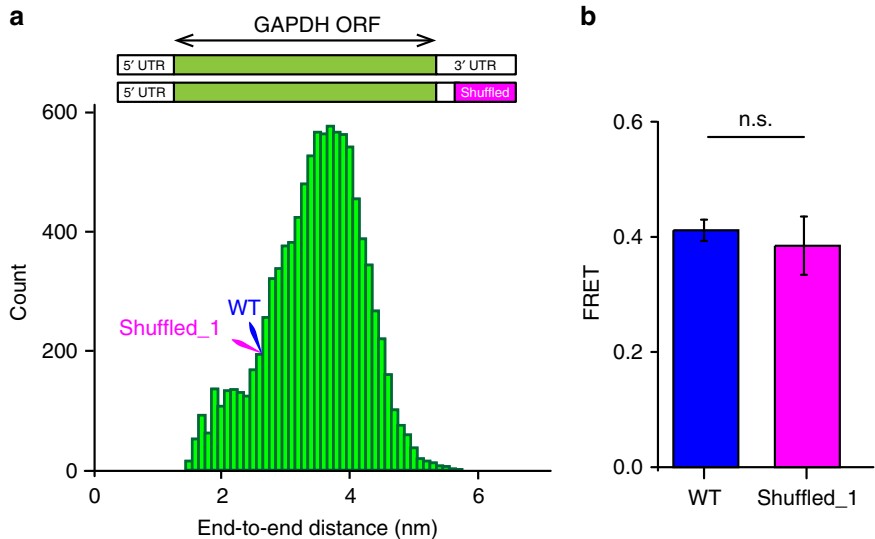

**Fig. 4** RNA end-to-end distance is largely sequence independent. **a** Distribution of average end-to-end distances in 10,000 GAPDH mRNA variants, in which 106 3′ terminal nucleotides were shuffled. Predicted end-to-end distances in wild-type and shuffled_1 GAPDH mRNAs are indicated by blue and pink arrows. **b** Ensemble FRET values measured in wild-type (blue) and shuffled_1 (pink) GAPDH mRNA variants. Each FRET value represents the mean ± SD of three independent experiments. The difference between FRET values was not statistically significant (n.s.) as determined by the Student t-test with $\alpha$ of 0.05

sequence in silico using a genetic algorithm. In this newly developed algorithm, populations of sequences are evolved either by random mutation or by crossover (combination of two sequences from the population), and then sequences with the lowest mean basepairing probabilities are selected for subsequent iterative refinement (details can be found in Methods). We performed 500 independent in silico evolution transformations of the GAPDH mRNA sequence; each of these transformations entailed 1000 mutation/sequence selection iterations. For each iteration of sequence evolution, we estimated RNA end-to-end distance and sequence linguistic complexity[40,41]. The latter quantity is bounded to be greater than zero and less than or equal to 1, where complexities reflect a larger diversity in oligonucleotide sequences within a sequence (the quantitative definition can be found in Methods).

In silico evolution transformations of the GAPDH mRNA sequence revealed that a reduction in average basepairing probability leads to an increase in end-to-end distance of RNA (Fig. 5a). In addition, the RNA sequence becomes enriched with

cytosines and depleted of guanosines (Fig. 5b). Guanosines are likely depleted because, in addition to Watson–Crick G–C base pairs, they can form wobble base pairs with uracils. G–U wobble base pairs have comparable thermodynamic stability to Watson–Crick A–U base pairs and are nearly isosteric to them[42,43].

A reduction in average basepairing probability is also accompanied by a decrease in sequence linguistic complexity (Fig. 5a) leading to the emergence of degenerate and repetitive sequences. The heat map showing alterations in the distribution of sequence complexities and end-to-end distances over 500 independent in silico evolution transformations of the GAPDH mRNA sequence indicates that the sequence complexity and end-to-end distance undergo anti-correlated changes (Fig. 5c).

Although the decrease in sequence complexity levels off and sequences become completely depleted of guanosines after ~200 iterations, basepairing probability and end-to-end distance continue evolving until they plateau after ~500 iterations. The maximal value of end-to-end distance (46 nm) achieved during in

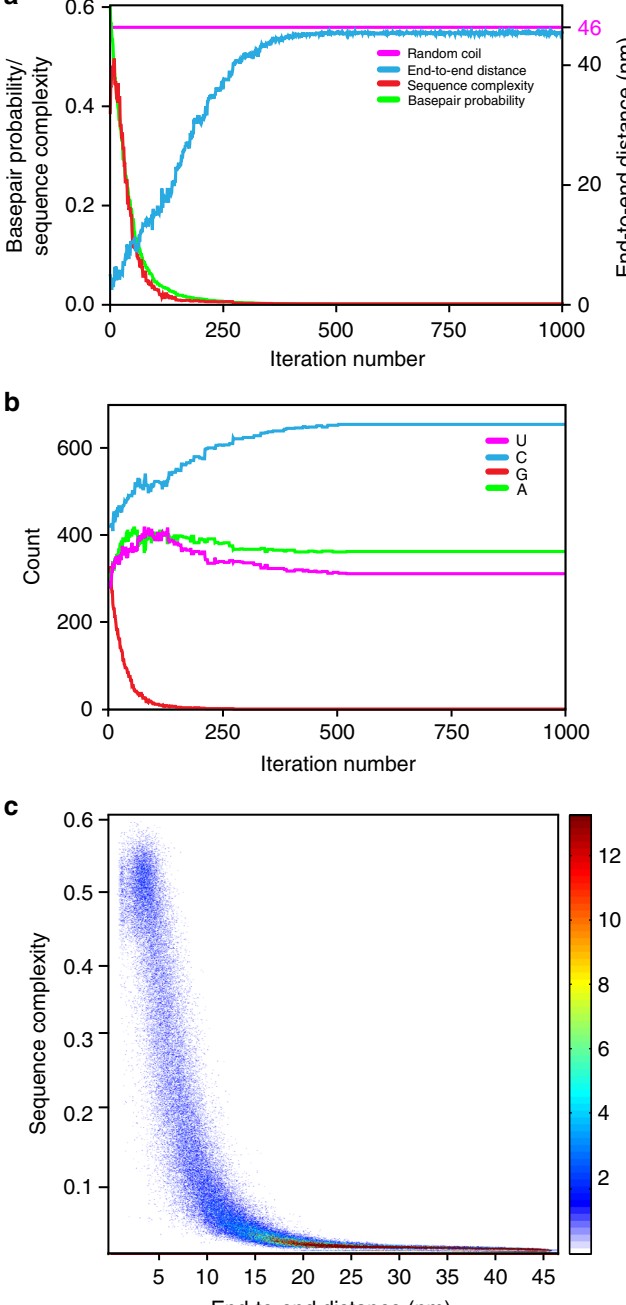

**Fig. 5** Sequence features of intrinsically unstructured RNA sequences. The entire 1327-nt-long GAPDH mRNA sequence was evolved in silico by a genetic algorithm to minimize average basepairing probability and produce intrinsically unstructured sequences. **a** End-to-end distance (blue), sequence complexity (red), and mean base pair probability (green) as functions of iteration number are shown for a single representative in silico sequence evolution experiment. The distance predicted for a 1327-nt-long RNA in a random-coil conformation is shown by the magenta line. **b** Evolution of nucleotide composition in a single representative in silico sequence transformation experiment shown in **a**. Frequency of adenosine (A), cytidine (C), guanosine (G), and uridine (U) are shown in magenta, blue, red, and green, respectively. **c** Surface contour plots generated from 500 independent in silico sequence evolution experiments show changes of sequence complexity (y-axis) as a function of end-to-end distance (x-axis). The range of sequence complexity from 0 to 0.6 was separated into 2000 bins. The range of end-to-end distance from 1.6 to 46 nm was separated into 500 bins. The resulting heat map shows the frequency count

silico sequence evolution of the 1327-nt long GAPDH mRNA is equal to the end-to-end distance predicted for RNA of this length in the random-coil conformation. Therefore, guanosine depletion and diminishing of sequence complexity are necessary but not sufficient to convert structured RNA into a completely unstructured conformation. Only specific, low-complexity sequences of adenosines, cytosines and uracils adopt the random-coil conformation. Hence, if intrinsically unstructured RNA sequences occur in organisms, then they likely have an important biological role and emerged as results of intense natural selection.

**Design of non-repetitive unstructured RNA sequences.** We and others find that long repetitive sequences, such as CA and CAA repeats, which are commonly introduced into RNA to disrupt RNA secondary structure, are notoriously difficult to maintain and propagate in live cells[44]. To overcome this problem, we employed our genetic algorithm to generate 500 sequences of human GAPDH mRNA, in which the last 106 nucleotides of the 3′ UTR were evolved into non-repetitive, intrinsically unstructured sequences (Fig. 6a, b). During sequence evolution, the selection criteria were changed to consider both linguistic complexity and mean basepairing probability to increase end-to-end distance and also avoid highly repetitive sequences. One of the resulting GAPDH mRNA sequences was cloned, transcribed without the 3′ poly(A) tail and fluorescently labeled for FRET measurements of end-to-end distance. Consistent with computational prediction, introduction of the non-repetitive, unstructured sequence into the 3′ end of the 3′ UTR of GAPDH mRNA led to a marked decrease in energy transfer between fluorophores attached to mRNA ends (Fig. 6c). This proof-of-principle experiment demonstrates that our new genetic algorithm for sequence evolution can help to design non-repetitive unstructured RNA sequences, which may be employed to study roles of RNA secondary structure in different aspects of RNA function.

## Discussion

Taken together, our data strongly support the hypothesis[9–11] that RNA as a polymer has an intrinsic propensity to fold into structures in which the 5′ and 3′ ends are just a few nm apart. Furthermore, we show that the ends of natural human mRNA and lncRNA sequences, folded in the absence of protein factors, are universally close. This occurs not only because of base pairs between nucleotides in the 5′ and 3′ UTRs but also because stem loop formation across whole sequences tends to shorten the end-to-end distance (Fig. 1a, Supplementary Fig. 1).

We find that RNA end-to-end distance is largely sequence independent. In contrast to tRNAs, rRNAs and some lncRNAs, mRNAs are unlikely under selective pressure to fold into a single specific secondary structure, in which the 5′ and 3′ ends make evolutionary-conserved basepairing interactions. Indeed, our computation and smFRET studies show that each mRNA sequence folds into a dynamic ensemble of structures with distinct but nevertheless short end-to-end distances. Hence, in the ensemble of structures, mRNA ends are brought in close proximity by a number of alternative helixes formed between the 5′ and 3′ UTRs rather than by one specific set of base pairs between mRNA ends. Our ensemble FRET measurements also show that the 5′ and 3′ ends of ~1700 nt-long firefly luciferase ORF, which lacked natural 5′ and 3′ UTRs, are just 5 nm apart from each other (Fig. 1b, Supplementary Table 1), further demonstrating that basepairing interactions between RNA ends do not require specific, evolutionary-conserved complementary UTR sequences.

At least to some degree, the intrinsic mRNA and lncRNA propensity of folding into structures with short end-to-end distances is likely realized in live cells. It is possible that co-

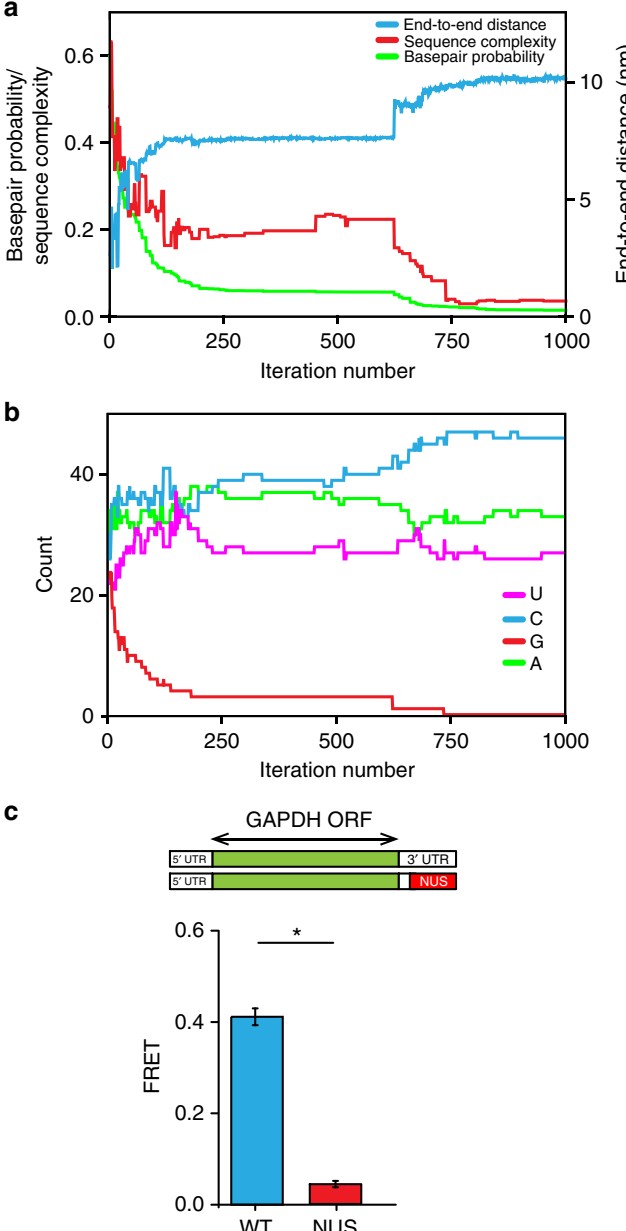

**Fig. 6** Manipulation of end-to-end distances in GAPDH mRNAs in silico with a genetic algorithm. 106 nucleotides in the 3′ end of the 3′ UTR in GAPDH mRNA are computationally evolved in silico by a genetic algorithm. **a** End-to-end distance (blue), sequence complexity (red), and base pair probability (green) as functions of iteration number are shown for a single in silico sequence evolution experiment. **b** Evolution of nucleotide composition in the sequence evolution in silico experiment shown in **a**. Frequency of adenosine (A), cytidine (C), guanosine (G), and uridine (U) are shown in magenta, blue, red, and green, respectively. **c** FRET values were measured in wild-type GAPDH mRNA (blue) and the GAPDH mRNA variant with the non-repetitive unstructured (NUS) 106 nucleotide sequence in the 3′ end of the 3′ UTR (red) designed by a genetic algorithm. Each FRET value represents the mean ± SD of three independent experiments. A star indicates that FRET values are different, as p-values determined by the Student t-test were below 0.05

transcriptional RNA folding in vivo might be different from the folding of full-length RNA in vitro in terms of both the folding pathway and final formed structure. Furthermore, RNA secondary structure may be disrupted by RNA-binding proteins and

RNA helicases[5,45]. In addition, the ribosome efficiently unwinds the secondary structure of mRNA within an open reading frame (ORF) by translocating along the mRNA during the elongation of the polypeptide chain[46,47]. However, RNA interactions with the ribosome and RNA-binding proteins are dynamic and, thus, at least transiently, mRNA and lncRNA likely fold in vivo. Indeed, several recent transcriptome-wide chemical probing studies in yeast, plant and human cells showed that mRNAs adopt extensive intramolecular secondary structure in vivo[45,48–51]. A number of structured elements within the mRNA, including bacterial riboswitches[52], frameshift-inducing hairpins and pseudoknots of eukaryotic viruses[53], Internal Ribosome Entry Sites (IRES)[54], Iron Response Elements (IRE) in the 5′ UTR of transcripts coding for proteins involved in iron metabolism[55], and Cap-Independent Translational Enhancers (CITEs)[56], were shown to regulate translation initiation. Studies have also shown that accessibility of protein binding sites on mRNA and mRNA polyadenylation are also governed by RNA secondary structure[5,57,58]. Hence, at least to some extent, the secondary structure of mRNA can be maintained in the cell despite the presence of RNA helicases and other RNA-binding proteins[5,49–51]. Moreover, experimental evidence suggests that the thermodynamics rather than kinetics of co-transcriptional folding appears to determine the predominant secondary RNA structures in live cells[59]. In addition, in vivo mapping of RNA–RNA interactions using psoralen cross-linking in yeast and human cells detected intramolecular basepairing interactions between distant segments of mRNA, including interactions between the 5′ and 3′ UTRs[49–51]. These data indicate that the 3′ end of the 3′ UTR may be brought near the 5′ end of mRNA through intramolecular basepairing interactions in live cells.

The inherent closeness of mRNA and lncRNAs ends may have important implications for many aspects of mRNA biology, including RNA decay and translation. For example, basepairing interactions between RNA ends may affect RNA susceptibility to degradation by exonucleases[6]. The intrinsic closeness of mRNA ends may also influence eukaryotic translation initiation, which involves protein-mediated interactions between mRNA ends. The recruitment of the small ribosomal subunit to the 5′ end of the mRNA during translation initiation is stimulated by the interaction between the 5′ mRNA cap-binding protein eIF4E and the 3′ poly(A) tail binding protein PABP, which is mediated through their binding to different parts of the translational factor eIF4G[60,61]. The eIF4E•eIF4G•PABP complex is thought to enhance translation initiation by circularizing the mRNA and forming the "closed-loop" structure[62–64]. The mechanism by which the mRNA closed loop enhances protein synthesis is not well understood.

Remarkably, translation initiation of many eukaryotic mRNAs is also regulated by sequences in their 3′ UTRs and controlled by the formation of protein bridges between the 5′ and 3′ UTRs. For example, the 3′ UTR regulatory sequences recruit protein complexes (e.g., CPEB•Maskin, Bruno•Cup, or GAIT complex), which inhibit translation by interacting with either eIF4E or eIF4E•eIF4G bound to the 5′ end of mRNA[65]. The pervasiveness of protein bridges between mRNA UTRs in the evolution of translation regulation is puzzling because of the significant entropic cost expected for protein-mediated mRNA circularization[10].

The entropic penalty for the formation of protein bridges between mRNA ends may be partially mitigated by mRNA compaction through intramolecular basepairing interactions. The protein complexes, which regulate translation by simultaneously interacting with the mRNA 5′ and 3′ ends or UTRs may have emerged during evolution to exploit the intrinsic closeness of mRNA ends. Thus, mRNA secondary structure, which brings

mRNA ends in close proximity, may stabilize the binding of translation factors bridging mRNA ends.

Spontaneous fluctuations of mRNA between different structural states, which were observed in our smFRET experiments, might also have a role in translation. The presence of stable secondary structure near the 5′ mRNA cap and the start codon was previously shown to inhibit translation initiation[7,8]. It is possible that during translation initiation, the 5′ end of the 5′ UTR undergoes partial unfolding while the rest of mRNA remains folded and compact, enabling recruitment of the small ribosomal subunit to the 5′ mRNA cap.

In the course of this work, we developed approaches and tools for measuring, computing, and manipulating mRNA end-to-end distances and the secondary structure of mRNA UTRs. This methodology can now be utilized to study unknown roles of mRNA end-to-end distance and secondary structure in mRNA UTRs in protein synthesis and other facets of mRNA biology.

## Methods

**RNA preparation.** To prepare mRNAs and lncRNAs, we employed run-off in vitro transcription catalyzed by 6-His-tagged T7 RNA polymerase. With the exception of the pcDNA3-FLUC plasmid containing the T3 promoter for transcription, all mRNA encoding sequences were cloned downstream of a T7 promoter. Cloning of RNA-encoding sequences is described in Supplementary Methods. To obtain the transcripts with or without a 30-nt poly(A) tail, plasmid DNAs were digested by EcoRI or SacI, respectively, and precipitated with ethanol. To obtain the rabbit β-globin transcript with or without a 30-nt poly(A) tail, plasmid DNAs were digested by AgeI or SacI, respectively. 10 μg of linearized plasmid DNA was added to a 1 mL transcription reaction mixture containing 80 mM HEPES-KOH pH 7.5, 2 mM spermidine, 30 mM dithiothreitol, 25 mM NaCl, 8 mM MgCl2, and 0.8 mM each of ATP, UTP, GTP, and CTP. The reaction was initiated by adding homemade T7 RNA polymerase to a final concentration of 2 μM and then incubated at 37 °C for 4–6 h. After the transcription reaction, the synthesized RNA was precipitated with 0.3 M sodium acetate pH 5.3 and 2.5 volumes of ethanol. The precipitated RNA was purified from a denaturing RNA gel (20 cm × 16 cm × 1.5 mm) in 7 M Urea, 1× TBE, and 5% acrylamide. The gel was pre-run at 20 mA for 30 min. An equal volume of formamide was added to the RNA before electrophoresis to aid denaturation of the RNA sample. RNA was run on the gel at 20 mA for at least 2 h until the tracking dye, Bromophenol blue, migrated three-fourths of the way through the gel. RNA bands were visualized on a TLC plate by briefly exposing the gel to short wave (254 nm) UV light. The RNA was eluted/recovered from the gel slab in gel extraction buffer (0.3 M sodium acetate pH 5.3, 0.5% SDS, and 5 mM EDTA) followed by phenol–chloroform extraction and ethanol precipitation.

**RNA labeling with Cy3 and Cy5 fluorophores.** The 5′ phosphate of the RNA was labeled using cystamine and a maleimide derivative of Cy5 (or Cy3, Click Chemistry) in the following steps as previously described[66]: (i) the 5′ γ-phosphate of RNA was reacted with 1-Ethyl-3-[3-dimethylaminopropyl]carbodiimide hydrochloride (EDC, Thermo Scientific) and imidazole; (ii) the phosphorimidazolide derivative of RNA was reacted with cystamine; (iii) the product was reduced with TCEP to release the thiophosphate group, which was subsequently modified by Cy5 (or Cy3) maleimide for 2 h at room temperature (RT). Labeled RNA molecules were purified using a 1 mL G-25 spin column equilibrated with ddH2O. We can reproducibly achieve 100% yield of 5′ phosphate labeling. The 5′-labeled RNAs were then subjected to 3′ end labeling. The RNA 3′ OH group was conjugated to pCp-Cy3 (or Cy5, Jena Bioscience) by T4 RNA ligase I in 20 mM MgCl2, 3.3 mM DTT, and 50 mM Tris-HCl pH 8.5 in the presence of 5 mM ATP and 10% DMSO. The efficiency of Cy3/5-pCp labeling of transcripts typically does not exceed 15–30%. To perform ensemble FRET experiments, the 5′ end of mRNAs was labeled with Cy3 to ensure that all acceptor (Cy5)-labeled RNAs were also labeled with a donor fluorophore. To perform single-molecule FRET experiments, the 5′ end of mRNAs was labeled with Cy5 to ensure that all donor (Cy3)-labeled RNAs, imaged using the green (532 nm) laser, were also labeled with an acceptor (Cy5) fluorophore. All labeled RNA molecules were gel purified by denaturing PAGE as described above and stored in ddH2O after desalting via a 1 mL G-25 spin column.

**RNA folding.** To measure the end-to-end distances of mRNAs and lncRNAs by ensemble FRET, 300 nM doubly labeled (5′-Cy3, 3′-Cy5) RNA samples were refolded in 30 μL of folding buffer (50 mM HEPES-KOH pH 7.5 and 100 mM KCl) by first heating RNA at 90 °C for 2 min and slowly cooling to 37 °C. At 37 °C, MgCl2 was added to a final concentration of 0.5 to 8 mM depending on the experiment and cooled to RT for 5 min. In most experiments, folding buffer contained 1 mM MgCl2. In the experiments shown in Supplementary Fig. 7, in addition to 1 mM MgCl2, PEG-8000 was added to a final concentration of 0–16%

(wt/vol). To disrupt the basepairing interactions between the 5′ and 3′ UTR in GAPDH or β-globin mRNAs (shown in Fig. 2), a 50-nucleotide-long DNA oligomer complementary to the 3′ end of the 3′ UTR of mRNA was added during mRNA refolding (GAPDH: 5′-CCTGGTTGAGCA-CAGGGTACTTTATTGATGGTACATGACAAGGTGCGGCT-3′ and β-globin: 5′-CGCAATGAAAATAAATTTCCTTTATTAGCCAGAAGTCA-GATGCTGAAGGG-3′).

For single-molecule experiments, 100 nM doubly labeled RNA samples (5′-Cy5, 3′-Cy3) and 150 nM biotinylated anchor DNA oligomers in 5 μL folding buffer were annealed as described above. Anchor DNA oligomers were designed to have a minimal effect on the overall secondary structure using OligoWalk software[23]: GAPDH: 5′-biotin/GATGATCTTGAGGCTGTTG-3′; β-globin: 5′-biotin/TAGGATTGTTCATAACAGCA-3′; and MIF: 5′-biotin/CATGTCGTAATAGTTGATGT-3′.

**Ensemble FRET measurements.** The fluorescence emission of Cy3 and Cy5 were measured using a FluoroMax-4 (Horiba) spectrofluorometer at RT. A 12.5 mm × 45 mm quartz cuvette with a 10 mm path length (Starna Cells) was used for a sample volume of 30 μl. All measurements were performed in folding buffer (50 mM HEPES-KOH pH 7.5 and 100 mM KCl) in the presence of 0.5 to 8 mM MgCl2 depending on the experiment. Cy3 emission spectra (555–800 nm) were taken by exciting fluorescence at 540 nm. Cy5 emission spectra (645–800 nm) were taken by exciting fluorescence at 635 nm. The slit-widths for excitation and emission were set to 5 nm of spectral bandwidth. FRET efficiencies (E) between the 5′ and 3′ end-labeled mRNAs were determined from Cy3 and Cy5 emission spectra using the ratioA method[67]. RatioA for each experiment was calculated from the ratio of the extracted integrated intensity of the acceptor (Cy5) fluorescence, which is excited both directly by 540 nm light and by energy transfer, divided by the integrated intensity of the acceptor excited directly by 635 nm light. FRET efficiency (E) was determined according to the following:

$$E = \left(\frac{1}{d^+}\right)\left(\frac{\varepsilon_A(635\,\text{nm})}{\varepsilon_D(540\,\text{nm})}\right)\left(\text{ratioA} - \frac{\varepsilon_A(540\,\text{nm})}{\varepsilon_A(635\,\text{nm})}\right) \qquad (1)$$

where $d^+$ is the fraction of molecules labeled with donor. The donor labeling efficiency was determined from absorbance spectra. $1/d^+$ indicates the fraction of acceptor-labeled RNA that is also labeled with donor by assuming that the labeling of each fluorophore was random. $\varepsilon$ is the extinction coefficient at the indicated wavelength, and the subscripts D and A denote donor or acceptor. The distance between donor and acceptor (R) was calculated from experimentally determined E values using:

$$E = \frac{1}{1 + \left(\frac{R}{R_0}\right)^6} \qquad (2)$$

where $R$ is the inter-dye distance and $R_0$ is the Förster radius at which $E = 0.5$. Calculations of R were performed assuming $R_0 = 56$ Å[68].

**Single-molecule FRET measurements.** 100 nM biotin-conjugated Cy3/Cy5 doubly labeled RNA samples were diluted in imaging buffer (50 mM HEPES pH 7.5, 100 mM KCl, 1 mM MgCl2, 0.625% glucose, and 1.5 mM Trolox) to a final concentration of 50 pM and immobilized on quartz slides coated with biotinylated BSA (0.2 mg/mL, Sigma) and pre-treated with NeutrAvidin (0.2 mg/mL, Thermo Scientific). Imaging buffer with an oxygen-scavenging system (0.8 mg/mL glucose oxidase and 0.02 mg/mL catalase) was injected into the slide chambers before imaging to prevent photobleaching.

smFRET traces were recorded using a prism-based total internal reflection fluorescence (TIRF) microscope as previously described[69,70]. The flow chamber was imaged using an Olympus IX71 inverted microscope equipped with a 532 nm laser (Spectra-Physics) and 642 nm laser (Spectra-Physics) for Cy3 and Cy5 excitation, respectively. Fluorescence emission was collected by a water immersion objective (×60/1.20 w, Olympus). Fluorescence signals were split into Cy3 and Cy5 channels by a 630 nm dichroic beam splitter and recorded by EMCCD camera (iXon+, Andor Technology). Movies were recorded using Single software (downloaded from Prof. Taekjip Ha's laboratory website at the Center for the Physics of Living Cells, University of Illinois at Urbana-Champaign (https://cplc.illinois.edu/software/))[13] with the time resolution of 0.1 s for 10 min. All experiments were performed at RT.

**Single-molecule data analysis.** Collected datasets were processed and trajectories for individual molecules were extracted with IDL, using scripts downloaded from https://cplc.illinois.edu/software/. Apparent FRET efficiencies ($E_{app}$) were calculated from the emission intensities of donor ($I_{Cy3}$) and acceptor ($I_{Cy5}$) as follows: $E_{app} = I_{Cy5}/[I_{Cy5} + I_{Cy3}]$. The FRET distribution histograms were built from more than 200 trajectories that showed single-step disappearance for both Cy3 and Cy5 fluorescence intensities using a Matlab script provided by Prof. Peter Cornish (University of Missouri, Columbia). Single-step photobleaching of the acceptor dye resulting in a reciprocal increase in donor fluorescence indicated the presence of an energy transfer before acceptor photobleaching. FRET histograms were fitted to

Gaussians using Origin (OriginLab). Among all collected traces, 20% of β-globin and GAPDH individual smFRET traces showed spontaneous interconversions between multiple FRET states. The state-to-state transitions in each fluctuating trace in GAPDH mRNA ($N = 266$) were determined using hidden Markov modeling (HMM) via HaMMy software[25]. smFRET traces showing apparent fluctuation were fit using HMM to 2, 3, and 4-state models. 99% of smFRET traces were best fit by 2-state models. Idealized FRET traces obtained by HMM were examined using transition density plot (TDP) analysis. To obtain TDP, the range of FRET efficiencies from 0 to 1 was separated into 200 bins. The resulting TDP heat map was normalized to the most populated bin in the plot. The lower-bound and upper-bound thresholds were set to 5% and 100% of the most populated bin, respectively.

**Estimating end-to-end distance by computation**. To estimate end-to-end distance distributions and mean end-to-end distances for each RNA sequence, we used a two-scale freely jointed chain approximation[19] for each structure in a Boltzmann ensemble of structures. We generate 1000 structures using stochastic sampling[34] (program stochastic) in RNAstructure[33]. In stochastic sampling, structures are selected at random with the probability equal to their Boltzmann probability. Because the sample is Boltzmann weighted, the mean of a quantity across the sample has the proper Boltzmann weighting. For each structure, we count the number of branches and unpaired nucleotides in the exterior loop, i.e., the loop that contains the 5′ and 3′ ends, and use:

$$D = \sqrt{a^2 n^{\frac{6}{5}} + b^2 m^{\frac{6}{5}}} \qquad (3)$$

where $D$ is the end-to-end distance, $n$ is the number of unpaired nucleotides, $m$ is the number of helical branches, $a = 6.2$ Å, and $b = 15$ Å, where $a$ and $b$ were from a previous parameterization[19]. The mean end-to-end distance is the arithmetic mean across structures in the sample. mRNA sequences of HeLa cells transcriptome were downloaded from the UCSC Genome Browser (September 22, 2017). Human lncRNA sequences were downloaded from the LNCipedia database (June 22, 2018)[36].

The genetic algorithm and computational analyses of the dependence of end-to-end distance on GC content and sequence complexity are described in Supplementary Methods.

**Code availability**. The genetic algorithm will be provided for free download as part of the RNAstructure software package (https://rna.urmc.rochester.edu/RNAstructure.html). This package is provided under the GPL V3 license, and is therefore open source and free.

## Data availability

The data that support the findings of this study are available from the corresponding author upon request.

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

## Acknowledgements

This study was supported by grant from the US National Institute of Health R01GM099719 (to D.N.E.) and R01GM076485 (to D.H.M). We thank Gloria Culver, Michael Sloma and Michael Cross for their early contributions to this project.

## Author contributions

D.N.E. conceived the project. W.-J.C.L., M.K., D.H.M. and D.N.E. designed research. W.-J.C.L. performed experiments with contributions from E.V.C., E.F., R.R. and E.S., M.K. performed computational studies with contributions from S.B., W.-J.C.L., M.K., D.H.M. and D.N.E. wrote the paper.

## Additional information

**Competing interests:** The authors declare no competing interests.

