## [Peer Review File · Nature Communications]

Reviewers' comments:

Reviewer #1 (Remarks to the Author):

Lai et al. present a combination of biophysical and computational analyses of the intrinsic propensity of mRNAs to assume conformations that bring the 5' and 3' ends into proximities on the order of a few nanometers. The authors extend upon several previously published studies investigating this same property of large RNAs by providing additional examples where the model holds up. The study describes what appears to be a novel computational method with which to analyze distributions of end-to-end distances at the transcriptome level. In addition, the authors report an interesting method for in silico evolution of non-repetitive intrinsically unstructured RNA sequences, which should be of use in the area of RNA design applications. While the study reported by Lai et al is certainly thought provoking, it appears to represent a somewhat incremental advance over pre-existing models of end-to-end distance properties of large RNAs. Moreover, although the authors do a commendable job comparing results of computational prediction with a few experimental tests, the paper lacks any direct test of the physiological significance of the main finding. Due to these limitations, this study is not likely to be appropriate for the broad readership of Nature Communications. Nevertheless, the authors may wish to address some of the following specific points in order to strengthen their study.

1. The authors should make a clear case for the limitations of related previous work and describe how their current study makes an important advance over earlier work. The authors test just a handful of the RNAs in the smFRET experiment, all of which agree with the computational expectation. However, it is not convincing that one should take these data to be representative of all mRNAs.

2. Perhaps the biggest limitation of the work as presented is the lack of any data relating their finding of close end-to-end distance with some functional output. In other words, the authors should consider leveraging what they have learned from the reported experiments to design an incisive experiment that demonstrates the functional link between end-to-end distance and some biological process. For example, can a modified 3'UTR be designed and put into a biological assay to test specific predictions of the impact caused by disruption of the close end-to-end distance? Conversely, can sequences be designed that enhance the stability of a close end-to-end distance conformation that in turn improves the translational efficiency of the RNA? Having some data of this kind would significantly enhance the impact of the study.

3. The authors discuss the importance of intramolecular base pairing within a large mRNA in bringing the ends close together; however, they do not appear to directly test this statement. Rather this claim is rooted in the results of computational RNAstructure predictions that show, as expected, base pairing potential throughout the RNAs. It would be interesting to see a test of a construct with native 5' and 3' UTR sequence but with an intrinsically unstructured internal connecting sequence. Such an experiment would directly test whether pairing between the ends is both necessary and sufficient to stabilize an RNA conformation (or ensemble of conformations) that achieves a close end-to-end distance.

4. In Figure 1, the authors may wish to show a secondary structure prediction model for which they are reporting data in the rest of the figure, rather than the MIF RNA. This would help the reader relate the sequence of the RNA (ie. Beta-globin) to the rest of the figure, for example, expected effect of disrupting base pairing with a 50-mer DNA oligo at the 3' end.

5. In general the experiments described in the paper are really focusing on the structural properties of the mRNA termini rather than the entire RNA as is modeled in the computational figures. To this reviewer, it would seem to be important to distinguish between the contributions of 'proximal' base

pairing versus 'distal' base pairing to the high FRET signal. The oligo disruption experiment is likely impacting the former, wherein nucleotides that are close in the primary sequence are folding into some structure. While it is expected that disruption of this local structure should impact the observed FRET value, it is not clear how critical this structure might be to the observed end-to-end distances that are the focus of the study. One could imagine that local folding may be necessary to permit tertiary interactions required to stabilize a close end-to-end distance, but this notion has not been explicitly tested. In a related point, the FRET dynamics the authors report are also likely to reflect 'local' changes in base pairing at the termini, which may or may not contribute directly to the close end-to-end distance property of the RNA. Of course these dynamics are interesting from the perspective of fundamental properties of RNA, but again, the functional significance these dynamics is not demonstrated in the current study.

Reviewer #2 (Remarks to the Author):

In the submitted manuscript the authors report the results of an extensive in silico and experimental analysis of the distance between mRNA ends. The propensity of mRNA to circularize and its implications to gene expression have been long recognized, however, an analysis of the factors that drive mRNA circularization have not been fully explored. This very interesting study provides insights into the roles of RNA base pairing to circularization and find that (primarily) non-specific base pairing within mRNAs is an important factor in circularization. The authors have done a wonderful job combining theoretical analyses with experimental validations and the manuscript is very well-written; such that the results are very clear to follow. My major concern is that the results of their work indicate that close end-to-end distances, rather than being a feature of mRNA, appear to be a general feature of RNA. It does not seem justified to focus in on mRNA and I recommend that the authors make revisions to broaden the scope of this manuscript.

Major Points:

1. Based on their results, circularization does not appear to be a special property of mRNAs, but rather a general feature of long RNA molecules: e.g. the authors show that randomized sequences behave similar to native mRNA sequences with respect to their end-to-end distances. The title of the paper could be changed to replace "mRNA" with "RNA" and more emphasis be placed on the propensity of RNAs in general to bring their ends close together in space.
2. One missing piece of this study is a comparison to protein folding. The distance between protein N and C termini has been described as being unusually low. This phenomenon was explained based on statistical arguments about the general properties of polymers (a "random flight of chains") [1]. Further work explored the importance of protein secondary structure in this observed close proximity of protein ends [2]. The claims of the unusual closeness of protein ends have, however, been challenged by additional models that find that the close proximity of ends is not lower than expected for random arrangements: though the authors admit that "closeness" can be a debated term [3]. A discussion comparing protein to RNA would greatly strengthen this manuscript: particularly with regards to end-to-end associations being potentially a general feature of polymers.
3. Related to the first point, the studies that disrupted base pairing potential of the UTRs and showed reductions in FRET efficiency; however, if the UTRs were simply clipped off and the donor and acceptor molecules placed before and after the start and stop codons of the coding sequence, it's likely that they would see similar results. It would seem that that mRNA UTRs are not important for circularization. I don't think this experimental control would be essential for publication, but, it would have been nice to include no UTR constructs in the FRET experiments. It would, however, be interesting to see how the calculated end-to-end distance plots would look comparing WT sequences to mRNAs with UTRs removed.

4. With so many results pointing to this being a general feature of longer RNAs, the authors should include rRNAs and long noncoding RNAs in the in silico analyses. It would be interesting to see how the lncRNA data compare to mRNAs with respect to the end-to-end distances. While not essential, FRET analyses of a highly structured ncRNA (e.g. rRNA) would be very interesting.
5. Another in silico study that could be very interesting would be to generate synthetic sequences spanning different nt contents to show how skews in sequence composition (e.g. GC%) could affect the end-to-end distances. A comparison of these results to natural mRNA and lncRNA sequence results would be very interesting.
6. A discussion of the potential roles of local kinetic folding would also be helpful. Both the in silico analyses and FRET studies fold the whole RNA transcript, which favors thermodynamically stable long-range interactions. In vivo, however, the growing RNA molecule folds, associates with proteins, and is processed co-transcriptionally. It's possible that local, kinetically favored, in vivo interactions might inhibit the formation of these energetically favored long-range pairs.
7. Another in vivo feature that would be exciting to consider are the effects of molecular crowding. Would a crowded environment be expected to favor or disfavor end-to-end associations in long RNAs? While not essential for publication, it would be interesting to see the effects of crowding agents on the FRET studies. For example, if the rate of the end-to-end association could be assayed by FRET using a range of crowding agent concentrations?

Minor Points:

1. 143-5 – the authors write that calculations predict mRNAs fold into an ensemble of structures. Do they mean calculations from this study or from the literature? Please clarify and give pointers (e.g. references) to the data.
2. A few grammatical errors were noticed in the paper. Nothing major, but another round of proof reading could be useful.
3. Very minor: Within the results section, there was a slight over use of the word "Hence", in my opinion. Variation in sentence openers may give the manuscript a better flow while reading.

References:

1. Flory PJ. Statistical Mechanics of Chain Molecules. New York: Wiley; 1969.
2. Thornton JM, Sibanda BL. Amino and carboxy-terminal regions in globular proteins. J Mol Biol. 1983;167(2):443-60.
3. Christopher JA, Baldwin TO. Implications of N and C-terminal proximity for protein folding. Journal of Molecular Biology. 1996;257(1):175-87.

Reviewer #3 (Remarks to the Author):

In this interesting and innovative study by Wan-Jung and colleagues, the authors present data that suggests that mRNAs have an intrinsic propensity to fold such that the end to end distance between the 5' and 3' end is less than 7 nm apart. This is much closer than what would be expected if the RNA acted as a random coil and suggests this is an intrinsic property of the RNA sequence. Furthermore, the authors also design computationally mRNAs with no propensity to base-pair between the 5' and 3' ends and show that these mRNAs have large end-end distances. This study elegantly combines smFRET measurements with structural modeling and addresses an interesting question. Although we know that in the cell the 5' cap and 3' UTR are brought in close proximity, the role of the mRNA in driving this proximity has not been shown convincingly until this study. As a result these data should be of interest to a broad readership. I have several comments the authors will need to address.

1.) I had trouble figuring out if any of the smFRET experiments were carried out in living cells. From the methods it appears all of these mRNAs were folded in vitro. This is in fact a strength of this study as it removes the contribution of endogenous proteins to any distance measurements. Nonetheless, the authors could make this more clear in the abstract, and perhaps rewrite it to focus less on the role of proteins in bringing the 5' and 3' ends of mRNAs together.

2.) In cells, mRNAs are often compartmentalized and may fold differently. For example, if they are translationally repressed their coding sequence is likely not actively unfolded by helices, but if they are actively being translated, it is likely the coding sequence won't fold. Is it possible to model, perhaps by constraining coding bases to be unpaired how that might affect the end to end distance? I would predict active translation might even bring the ends closer.

3.) If we hypothesize that end to end proximity is a way to more efficiently recycle ribosomes translating, might it not also be interesting to measure in the computational models the end to end distance of the start and stop codons? Furthermore, does this at all correlate with metrics of translation efficiency measured by ribosome profiling?

4.) Similarly, have the authors considered if simply end to end distance using their model is predictor of translation efficiency? Alternatively one might think this could also correlate with stability by somehow protecting the mRNAs from nucleases?

In summary, I believe that a couple more simple correlation analyses between the end to end distances computed on the transcriptome and existing data sets could further improve the biological impact of the work.

We are submitting a revised manuscript titled “mRNAs and lncRNAs intrinsically form secondary structures with short end-to-end distances”. We thank reviewers for their comments. Following the reviewers’ critiques and suggestions, we made multiple changes to the manuscript (highlighted in red), added a substantial amount of new data and expanded the scope of the manuscript. New experimental data are now included in several figures (Fig. 1b, Fig. 3b-c, Suppl. Figures 2, 7 and 9) and in Suppl. Table 1. We also revised Fig. 1a and Suppl. Fig. 1. Following the suggestions of reviewer 2, we used computation and FRET experiments to show that similarly to mRNAs, the ends of lncRNAs are universally close (Fig. 3b-c). We changed the title of our manuscript to reflect this important new finding. We hope that the revised manuscript will be suitable for publication in *Nature Communications*. Our detailed responses to the reviewers’ comments are below:

Response to reviewers’ comments:

Reviewer #1 (Remarks to the Author):

Comment: Lai et al. present a combination of biophysical and computational analyses of the intrinsic propensity of mRNAs to assume conformations that bring the 5’ and 3’ ends into proximities on the order of a few nanometers. The authors extend upon several previously published studies investigating this same property of large RNAs by providing additional examples where the model holds up. The study describes what appears to be a novel computational method with which to analyze distributions of end-to-end distances at the transcriptome level. In addition, the authors report an interesting method for in silico evolution of non-repetitive intrinsically unstructured RNA sequences, which should be of use in the area of RNA design applications. While the study reported by Lai et al is certainly thought provoking, it appears to represent a somewhat incremental advance over pre-existing models of end-to-end distance properties of large RNAs. Moreover, although the authors do a commendable job comparing results of computational prediction with a few experimental tests, the paper lacks any direct test of the physiological significance of the main finding. Due to these limitations, this study is not likely to be appropriate for the broad readership of *Nature Communications*. Nevertheless, the authors may wish to address some of the following specific points in order to strengthen their study.

1. The authors should make a clear case for the limitations of related previous work and describe how their current study makes an important advance over earlier work. The authors test just a handful of the RNAs in the smFRET experiment, all of which agree with the computational expectation. However, it is not convincing that one should take these data to be representative of all mRNAs.

Response:

We believe that our studies made several important advances over earlier work. (i) Previous computational studies explored end-to-end distances in randomized RNA libraries and in a limited number of natural sequences, while a single published FRET study examined end-to-end distance in several fungal and viral transcripts. By contrast, using computation, we investigated end-to-end distance in all human mRNAs. Using FRET, we measured end-to-end distances in a number of human mRNAs. For the revision, we also measured computationally and experimentally end-to-end distances for lncRNAs. Thus, our studies provide a crucial link between theoretical studies of RNA folding and biologically important human RNA molecules. (ii) In a published smFRET study aimed to test the hypothesis about the intrinsic closeness of mRNA ends, end-to-end distance was measured only in molecules that showed FRET, which might have represented only a minor fraction of the total population of RNA molecules. By contrast, because we measured ensemble FRET in addition to single-molecule FRET, we examined average end-to-end distance in tested mRNA molecules. Thus, our study presents the first rigorous experimental test of the hypothesis about the intrinsic closeness of mRNA ends. We revised the text to explain this contrast in the experimental methods used (page 3). Furthermore, because the previous FRET study measured energy transfer between mRNA ends in freely diffusing molecules, no insight into mRNA structural dynamics was gained. By contrast, we immobilized mRNA molecules in smFRET experiments and, thus, observed RNA structural dynamics on a seconds-to-minutes time scale. We found that mRNA molecules fold into an ensemble of interconverting structures with different end-to-end distances. (iii) Our work provides the first

experimental evidence that mRNA ends are brought in close proximity by basepairing interactions (a previously published FRET study did not test the effect of intramolecular basepairing interactions on end-to-end distances). (iv) Our computation and FRET studies were performed for the same transcripts. Thus, using FRET, we could validate computational modeling of end-to-end distances in RNA. The agreement between predicted and experimentally measured changes in end-to-end distance of computationally-designed sequences further demonstrates that our model can be generalized and applied to all RNA sequences. (v) Computational tools developed in our work enable rational design and manipulation of secondary structure and end-to-end distances in RNA. These computational tools may be used in a wide range of applications. We elaborated on these points in the revised manuscript. In sum, we believe that our study provides a number of important novel insights into universal intrinsic properties of RNA and will be quite interesting for the broad readership of *Nature Communications*.

Comment: 2. Perhaps the biggest limitation of the work as presented is the lack of any data relating their finding of close end-to-end distance with some functional output. In other words, the authors should consider leveraging what they have learned from the reported experiments to design an incisive experiment that demonstrates the functional link between end-to-end distance and some biological process. For example, can a modified 3'UTR be designed and put into a biological assay to test specific predictions of the impact caused by disruption of the close end-to-end distance? Conversely, can sequences be designed that enhance the stability of a close end-to-end distance conformation that in turn improves the translational efficiency of the RNA? Having some data of this kind would significantly enhance the impact of the study.

Response: We agree with the reviewer that investigating biological implications of the intrinsic closeness of mRNA ends is the next important and logical step in our studies. We are currently testing the effects of intramolecular basepairing interactions between the 5' and 3' UTRs on translation initiation in eukaryotes. Our preliminary data show, consistent with our hypothesis discussed in our manuscript, that the replacement of the original 3'UTR sequence in model mRNAs with unstructured sequences increases the intrinsic end-to-end mRNA distance and significantly hampers translation efficiency both in cell lysates and *in vivo*. However, further comprehensive and rigorous analysis of the effects of the mRNA end-to-end distance on translation is required before these results can be published. Such extensive work is clearly beyond the scope of the current manuscript.

Comment: 3. The authors discuss the importance of intramolecular base pairing within a large mRNA in bringing the ends close together; however, they do not appear to directly test this statement. Rather this claim is rooted in the results of computational RNAstructure predictions that show, as expected, base pairing potential throughout the RNAs. It would be interesting to see a test of a construct with native 5' and 3' UTR sequence but with an intrinsically unstructured internal connecting sequence. Such an experiment would directly test whether pairing between the ends is both necessary and sufficient to stabilize an RNA conformation (or ensemble of conformations) that achieves a close end-to-end distance.

Response: We experimentally demonstrate that the disruption in basepairing between mRNA ends by either the introduction of unstructured sequences or annealing of a DNA oligo to the 5' or 3' end of mRNA dramatically increases end-to-end distance (Fig. 1c). Based on these experimental results and computational modeling of mRNA structure, we conclude that mRNA ends are brought in close proximity by the basepairing interactions. We also state in the manuscript: "This occurs not only because of base pairs between nucleotides in the 5' and 3' UTRs but also because stem loop formation across whole sequences tends to shorten the end-to-end distance." In other words, folding of the entire mRNA sequence contributes to the stabilization of mRNA structures with short end-to-end distances. Following the reviewer's suggestion, we performed new computational analysis in which we replaced the entire ORF of human GAPDH mRNA with CA repeats while leaving the 5' and 3' UTRs intact. Consistent with our hypothesis that the entire structure contributes to the short end-to-end distance, the computational analysis shows that this change of the ORF to a non-base pairing sequence does not significantly affect the predicted end-to-end distance. However, thermodynamic stabilities of this computationally predicted structure decreases by from -484.6 kcal/mol to -89.6 kcal/mol, indicating

that ORF folding contributes to RNA compaction and stabilization of RNA structures with short end-to-end distances. We could not test these predictions experimentally because it is extremely difficult to clone and maintain in recombinant vectors long, repetitive, low-complexity sequences equivalent in length to the ORF of GAPDH (1008 nt) or other model mRNA.

Comment: 4. In Figure 1, the authors may wish to show a secondary structure prediction model for which they are reporting data in the rest of the figure, rather than the MIF RNA. This would help the reader relate the sequence of the RNA (i.e. Beta-globin) to the rest of the figure, for example, expected effect of disrupting base pairing with a 50-mer DNA oligo at the 3' end.

Response: This is an excellent suggestion. In the revised manuscript, we replaced the panel in Figure 1 to show the secondary structure of beta-globin mRNA (Fig. 1a). We moved the MIF structure to the supplement (Suppl. Fig. 1).

Comment: 5. In general the experiments described in the paper are really focusing on the structural properties of the mRNA termini rather than the entire RNA as is modeled in the computational figures. To this reviewer, it would seem to be important to distinguish between the contributions of 'proximal' base pairing versus 'distal' base pairing to the high FRET signal. The oligo disruption experiment is likely impacting the former, wherein nucleotides that are close in the primary sequence are folding into some structure. While it is expected that disruption of this local structure should impact the observed FRET value, it is not clear how critical this structure might be to the observed end-to-end distances that are the focus of the study. One could imagine that local folding may be necessary to permit tertiary interactions required to stabilize a close end-to-end distance, but this notion has not been explicitly tested. In a related point, the FRET dynamics the authors report are also likely to reflect 'local' changes in base pairing at the termini, which may or may not contribute directly to the close end-to-end distance property of the RNA. Of course these dynamics are interesting from the perspective of fundamental properties of RNA, but again, the functional significance these dynamics is not demonstrated in the current study.

Response: (i) To address this, we added a new Suppl. figure (Suppl. Fig. 2) to the revised manuscript. It demonstrates results of computational analysis suggesting that annealing the long (50 nt) DNA oligo to the 3' end of the 3' UTR or the replacement of 106 nt of the terminal segment of the 3' UTR with CA repeats disrupts both proximal and distal (long-range) basepairing interactions. Both local and long-range basepairing interactions (e.g. basepairing interactions between the 5' and 3' UTRs) likely contribute to compacting RNA structure and bringing RNA ends in close proximity. (ii) Weak dependence of FRET between fluorophores attached to RNA ends on the concentration of magnesium ions suggests that tertiary interactions play a minor role (if any) in bringing RNA ends in close proximity (Suppl. Figures 5 and 6). (iii) Although we did not study the biological role of mRNA spontaneous structural dynamics, we hypothesize in the Discussion of our manuscript that "Spontaneous fluctuations of mRNA between different structural states, which were observed in our smFRET experiments, might also play a role in translation. The presence of stable secondary structure near the 5' mRNA cap and the start codon was previously shown to inhibit translation initiation. It is possible that during translation initiation, the 5' end of the 5' UTR undergoes partial unfolding while the rest of the mRNA remains folded and compact, enabling recruitment of the small ribosomal subunit to the 5' mRNA cap."

Reviewer #2 (Remarks to the Author):

Comment: In the submitted manuscript the authors report the results of an extensive in silico and experimental analysis of the distance between mRNA ends. The propensity of mRNA to circularize and its implications to gene expression have been long recognized, however, an analysis of the factors that drive mRNA circularization have not been fully explored. This very interesting study provides insights into the roles of RNA base pairing to circularization and find that (primarily) non-specific base pairing within mRNAs is an important factor in circularization. The authors have done a wonderful job combining theoretical analyses with experimental validations and the manuscript is very well-written; such that the results are very clear to follow. My major concern is that the results of their work indicate that close end-to-end distances, rather than being a feature of mRNA, appear to be a general feature of RNA. It does not seem justified to

focus in on mRNA and I recommend that the authors make revisions to broaden the scope of this manuscript. Based on their results, circularization does not appear to be a special property of mRNAs, but rather a general feature of long RNA molecules: e.g. the authors show that randomized sequences behave similar to native mRNA sequences with respect to their end-to-end distances. The title of the paper could be changed to replace “mRNA” with “RNA” and more emphasis be placed on the propensity of RNAs in general to bring their ends close together in space.

Response: We agree with the reviewer that most, if not all, RNA sequences have the propensity to fold into structures with short end-to-end distances. Nevertheless, our study specifically focuses on natural mRNAs and now also lncRNA sequences because of their biological importance. We would like to emphasize this to the broad readership of *Nature Communications* in the title of our article. It seems important because our comprehensive analysis of human mRNAs and lncRNAs sets our work apart from previous computational studies that examined the intrinsic closeness of RNA ends and analyzed the distribution of end-to-end distances in either randomized RNA sequences or in a limited number of natural sequences. Another reason to focus on mRNA and lncRNA is that closeness of the 5' and 3' ends in tRNAs and rRNAs was known for quite some time since determination of both secondary and 3-D structures for these classes of RNA molecules. Unlike tRNAs or rRNAs, however, mRNAs are unlikely to be under selective pressure to fold into a specific, unique secondary structure. Thus, our finding that the ends of mRNAs and lncRNAs are always in close proximity is particularly intriguing, novel and interesting.

Comment: 2. One missing piece of this study is a comparison to protein folding. The distance between protein N and C termini has been described as being unusually low. This phenomenon was explained based on statistical arguments about the general properties of polymers (a “random flight of chains”) [1]. Further work explored the importance of protein secondary structure in this observed close proximity of protein ends [2]. The claims of the unusual closeness of protein ends have, however, been challenged by additional models that find that the close proximity of ends is not lower than expected for random arrangements: though the authors admit that “closeness” can be a debated term [3]. A discussion comparing protein to RNA would greatly strengthen this manuscript: particularly with regards to end-to-end associations being potentially a general feature of polymers.

Response: We considered discussing end-to-end distances of proteins. However, because “closeness of the ends” has somewhat different meaning in the case of proteins and RNAs, we chose not to compare proteins and RNA in our manuscript. Most proteins fold into compact globular structures. In the references mentioned by the reviewer, the distance between the N and C termini is discussed in regard to dimensions of folded proteins (e.g. radius of gyration). In this context, the closeness of protein termini implies that the distance between N and C termini is significantly shorter than the distance expected based on chance and dimensions of a given protein. Also, there is no agreement between different papers on whether protein termini are generally closer than expected by chance or not. In contrast to proteins, published computational studies of RNA structure suggest that mRNAs and lncRNAs do not fold into globular structures (Seetin & Mathews, *J Comput Chem.* 2011, 32, 2232-44; Yoffe et al, *Proc Natl Acad Sci U S A.* 2008, 105, 16153-8). We and others compare the end-to-end distance in folded RNA with the end-to-end distance expected for the random coil conformation of RNA. In this context, the closeness of mRNA (or lncRNA) ends means that the distance between the 5' and 3' ends is much shorter than the distance expected for the random coil conformation of the sequence.

Comment: 3. Related to the first point, the studies that disrupted base pairing potential of the UTRs and showed reductions in FRET efficiency; however, if the UTRs were simply clipped off and the donor and acceptor molecules placed before and after the start and stop codons of the coding sequence, it's likely that they would see similar results. It would seem that that mRNA UTRs are not important for circularization. I don't think this experimental control would be essential for publication, but, it would have been nice to include no UTR constructs in the FRET experiments. It would, however, be interesting to see how the calculated end-to-end distance plots would look comparing WT sequences to mRNAs with UTRs removed.

Response: Using FRET, we showed that the 5' and 3' ends of ~1700 nt-long firefly luciferase ORF, which lacked natural 5' and 3' UTRs, were 5 nm from each other (Fig 1b, Suppl. Table 1). Hence, as correctly anticipated by the reviewer,

basepairing interactions between RNA ends do not require specific UTR sequences. We revised the manuscript to state this in the Discussion.

Comment: 4. With so many results pointing to this being a general feature of longer RNAs, the authors should include rRNAs and long noncoding RNAs in the *in silico* analyses. It would be interesting to see how the lncRNA data compare to mRNAs with respect to the end-to-end distances. While not essential, FRET analyses of a highly structured ncRNA (e.g. rRNA) would be very interesting.

Response: As suggested by the reviewer, we now performed computational analysis on the distribution of end-to-end distances in human lncRNAs and found that, similarly to mRNA, the ends of lncRNAs are universally close. These results are now included in Fig. 3 of the revised manuscript. Furthermore, we used FRET to experimentally measure the end-to-end distance in two functionally important human lncRNAs (HOTAIR and NEAT1_S), which are 2148 and 3734 nt-long, respectively. We found that the ends of HOTAIR and NEAT1_S lncRNAs folded *in vitro* are ~7 and 6 nm apart, respectively (Fig. 3 and Suppl. Table 1). These distances are 10 and 20-fold shorter than respective end-to-end distances expected for RNAs of the same length in the random coil conformation. These results suggest that closeness of the ends is a general property of both mRNAs and lncRNAs. We included these data in the revised manuscript and changed the title of our manuscript to reflect these new findings.

Comment: 5. Another *in silico* study that could be very interesting would be to generate synthetic sequences spanning different nt contents to show how skews in sequence composition (e.g. GC%) could affect the end-to-end distances. A comparison of these results to natural mRNA and lncRNA sequence results would be very interesting.

Response: We performed this analysis in response to this suggestion. Our computational analysis shows that variation in GC content has no significant effect on the end-to-end distance in RNA. The complete depletion of guanosines is necessary but not sufficient for obtaining intrinsically unstructured RNA sequences with long end-to-end distances. As suggested by the reviewer, we clarify this point by now including a Suppl. figure (Suppl. Figure 9) summarizing the results of this computational analysis in the revised manuscript.

Comment: 6. A discussion of the potential roles of local kinetic folding would also be helpful. Both the *in silico* analyses and FRET studies fold the whole RNA transcript, which favors thermodynamically stable long-range interactions. *In vivo*, however, the growing RNA molecule folds, associates with proteins, and is processed co-transcriptionally. It's possible that local, kinetically favored, *in vivo* interactions might inhibit the formation of these energetically favored long-range pairs.

Response: It is possible that co-transcriptional RNA folding *in vivo* might be different from the folding of full-length RNA *in vitro* in terms of both the folding pathway and final formed structure. However, to what extent and how RNA folding is affected by transcription kinetics is not fully understood. Studies of the Fedor lab suggested that the thermodynamics (and not kinetics of co-transcriptional folding) determine the predominant RNA structures in live cells (Mahen et al., Mol Cell 2005). In the revised manuscript, we now discuss the kinetic and thermodynamic aspects of RNA folding and cite the article by Mahen et al.

Comment: 7. Another *in vivo* feature that would be exciting to consider are the effects of molecular crowding. Would a crowded environment be expected to favor or disfavor end-to-end associations in long RNAs? While not essential for publication, it would be interesting to see the effects of crowding agents on the FRET studies. For example, if the rate of the end-to-end association could be assayed by FRET using a range of crowding agent concentrations?

Response: Following this suggestion, we examined the effect of a molecular crowder (PEG 8000) on end-to-end distances in GAPDH, β -globin and MIF mRNAs using ensemble FRET. The average end-to-end distance in GAPDH mRNA was not affected by PEG8000 (Suppl. Fig. 7). Addition of 8-16% PEG8000 to β -globin and MIF mRNAs produced a relatively small but nevertheless appreciable increase in FRET between RNA ends indicating the stabilization of mRNA structures with shorter end-to-end distances (Suppl. Fig. 7). These results are consistent with the idea that molecular

crowding promotes RNA folding and the formation of more compact RNA conformations. The relatively small effect of crowding on the end-to-end distance is consistent with computational predictions, indicating that while mRNAs form extensive secondary structure and become more compact they do not fold into globular, highly-condensed structures. The discussion of these results and a Suppl. figure are added to revised manuscript.

Comment: Minor Points. 1. 143-5 – the authors write that calculations predict mRNAs fold into an ensemble of structures. Do they mean calculations from this study or from the literature? Please clarify and give pointers (e.g. references) to the data.

Response: We clarified our statement and added appropriate references (“Published studies and our own computational predictions suggest that mRNAs fold into an ensemble of structures with comparable thermodynamic stabilities rather than a single structure”).

Reviewer #3 (Remarks to the Author):

Comment: 1) I had trouble figuring out if any of the smFRET experiments were carried out in living cells. From the methods it appears all of these mRNAs were folded *in vitro*. This is in fact a strength of this study as it removes the contribution of endogenous proteins to any distance measurements. Nonetheless, the authors could make this more clear in the abstract, and perhaps rewrite it to focus less on the role of proteins in bringing the 5' and 3' ends of mRNAs together.

Response: We clarified in the abstract and throughout the manuscript that mRNAs and lncRNAs investigated in our studies by FRET were folded *in vitro* in the absence of proteins. We also moved the discussion of translational factors that regulate translation by bridging mRNA ends from Introduction to Discussion of the revised manuscript.

Comment: 2) In cells, mRNAs are often compartmentalized and may fold differently. For example, if they are translationally repressed their coding sequence is likely not actively unfolded by helices, but if they are actively being translated, it is likely the coding sequence won't fold. Is it possible to model, perhaps by constraining coding bases to be unpaired how that might affect the end to end distance? I would predict active translation might even bring the ends closer.

Response: As we mention above, our new computational analysis shows that replacing the entire ORF of human GAPDH mRNA with CA repeats while leaving the 5' and 3' UTRs intact does not significantly affect the predicted end-to-end distance. However, thermodynamic stabilities of computationally predicted structures decrease from -484.6 kcal/mol to -89.6 kcal/mol, indicating that ORF folding contributes to stabilization of RNA structures with short end-to-end distances. Hence, upon unfolding of the ORF by translating ribosomes the 5' and 3' UTRs might remain basepaired. However, the stability of mRNA structures with short end-to-end distances should decrease upon translation of the ORF.

Comment: 3) If we hypothesize that end to end proximity is a way to more efficiently recycle ribosomes translating, might it not also be interesting to measure in the computational models the end to end distance of the start and stop codons? Furthermore, does this at all correlate with metrics of translation efficiency measured by ribosome profiling? 4) Similarly, have the authors considered if simply end to end distance using their model is predictor of translation efficiency? Alternatively one might think this could also correlate with stability by somehow protecting the mRNAs from nucleases?

Response: We agree with the reviewer that one consequence of the intrinsic compactness of mRNA structure might be the closeness of the start and stop codons. However, estimating the distance between the start and stop codon is not straightforward and is beyond the scope of this manuscript. Translational efficiency measured by ribosome profiling is highly variable and can differ by at least two orders of magnitude. Our preliminary analysis, not included in the manuscript, suggests that translational efficiency does not correlate with predicted end-to-end distances in mRNAs. That is hardly surprising because our data indicate that end-to-end distances in mRNAs are nearly constant and do not

significantly vary between different transcripts (Fig. 1B). Nevertheless, as we hypothesize in the manuscript, the intrinsic closeness of the ends may enhance the recruitment of initiation factors and facilitate translation initiation of all transcripts. In addition, as suggested by the reviewer, we also mention in the revised manuscript that basepairing interactions between and within the 5' and 3' UTRs might affect mRNA stability and susceptibility to RNA degradation machinery.

REVIEWERS' COMMENTS:

Reviewer #1 (Remarks to the Author):

The authors have done a commendable job addressing the reviewer's concerns, primarily by revising text and performing additional computational experiments. The addition of lncRNAs in response to one reviewer certainly broadens the scope of the article and the authors do a nice job articulating how this work represents an important advance beyond previously published work. Since the RNA behavior they have characterized appears to be general ('universal' is a strong word), and the computational tools are likely to be useful for the broader RNA community, I now agree that the work may be of interest to the readership of Nature Communications.

Nevertheless, I still feel that some link between the RNA folding properties described in the paper and a biological or mechanistic function would significantly elevate the work. However, as the authors state, perhaps this would be beyond the scope of this manuscript. In light of this point, and the other reviewer comments, I recommend the paper be accepted for publication in its current form.

Reviewer #2 (Remarks to the Author):

The authors have gone above and beyond in their response to my reviews. I am glad that some of my suggestions appeared to be helpful and have no hesitations in recommending that this manuscript be accepted for publication in its current form.

Reviewer #3 (Remarks to the Author):

The authors have very nicely addressed all my comments and the paper can be published as is.